# On Linear Identifiability of Learned Representations

## Abstract

Identifiability is a desirable property of a statistical model: it implies that the true model parameters may be estimated to any desired precision, given sufficient computational resources and data. We study identifiability in the context of representation learning: discovering nonlinear data representations that are optimal with respect to some downstream task. When parameterized as deep neural networks, such representation functions lack identifiability in parameter space, because they are overparameterized by design. In this paper, building on recent advances in nonlinear Independent Components Analysis, we aim to rehabilitate identifiability by showing that a large family of discriminative models are in fact identifiable in *function* space, up to a linear indeterminacy. Many models for representation learning in a wide variety of domains have been identifiable in this sense, including text, images and audio, state-of-the-art at time of publication. We derive sufficient conditions for linear identifiability and provide empirical support for the result on both simulated and real-world data.

## 1 Introduction

An increasingly common methodology in machine learning is to improve performance on a primary down-stream task by first learning a high-dimensional representation of the data on a related, proxy task. In this paradigm, training a model reduces to fine-tuning the learned representations for optimal performance on a particular sub-task (Erhan et al., 2010). Deep neural networks (DNNs), as flexible function approximators, have been surprisingly successful in discovering effective high-dimensional representations for use in downstream tasks such as image classification (Sharif Razavian et al., 2014), text generation (Radford et al., 2018; Devlin et al., 2018), and sequential decision making (Oord et al., 2018).

When learning representations for downstream tasks, it would be useful if the representations were reproducible, in the sense that every time a network relearns the representation function on the same data distribution, they were approximately the same, regardless of small deviations in the initialization of the parameters or the optimization procedure. In some applications, such as learning real-world causal relationships from data, such reproducible learned representations are crucial for accurate and robust inference (Johansson et al., 2016; Louizos et al., 2017). A rigorous way to achieve reproducibility is to choose a model whose representation function is *identifiable* in function space. Informally speaking, identifiability in function space is achieved when, in the limit of infinite data, there exists a single, global optimum in function space. Interestingly, Figure 1 exhibits learned representation functions that appear to be the same up to a linear transformation, even on finite data and optimized without convergence guarantees (see Appendix A.1 for training details).

In this paper, we account for Figure 1 by making precise the relationship it exemplifies. We prove that a large class of discriminative and autoregressive models are identifiable in function space, up to a linear transformation. Our results extend recent advances in the theory of nonlinear Independent Components Analysis (ICA), which have recently provided strong identifiability results for generative models of data (Hyvärinen et al., 2018; Khemakhem et al., 2019; 2020; Sorrenson et al., 2020). Our key contribution is to bridge the gap between these results and *discriminative* models, commonly used for representation learning (e.g., (Hénaff et al., 2019; Brown et al., 2020)).

The rest of the paper is organized as follows. In Section 2, we describe a general discriminative model family, defined by its canonical mathematical form, which generalizes many supervised, self-

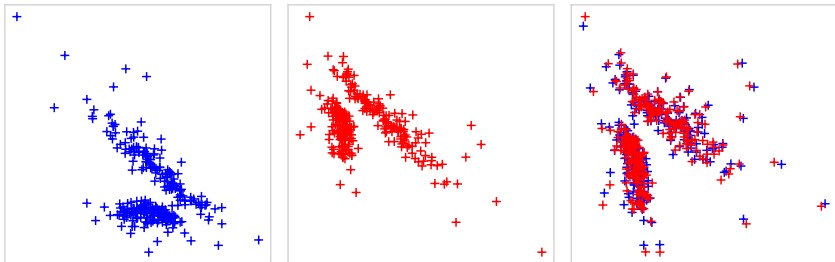

Figure 1: Left and Middle: Two learned DNN representation functions $\mathbf{f}_{\boldsymbol{\theta}_1}(\mathcal{B})$, $\mathbf{f}_{\boldsymbol{\theta}_2}(\mathcal{B})$ visualized on held-out data $\mathcal{B}$. The DNNs are word embedding models Mnih and Teh (2012) trained on the Billion Word Dataset (Chelba et al., 2013) (see Appendix A.1 for code release and training details). Right: $\mathbf{A}\mathbf{f}_{\boldsymbol{\theta}_1}(\mathcal{B})$ and $\mathbf{f}_{\boldsymbol{\theta}_2}(\mathcal{B})$, where $\mathbf{A}$ is a linear transformation learned after training. The overlap exhibits *linear identifiability* (see Section 3): different representation functions, learned on the same data distribution, live within linear transformations of each other in function space.

supervised, and contrastive learning frameworks. In Section 3, we prove that learned representations in this family have an asymptotic property desirable for representation learning: equality up to a linear transformation. In Section 4, we show that this family includes a number of highly performant models, state-of-the-art at publication for their problem domains, including CPC (Oord et al., 2018), BERT (Devlin et al., 2018), and GPT-2 and GPT-3 (Radford et al., 2018; 2019; Brown et al., 2020). Section 5 investigates the actually realizable regime of *finite* data and *partial* optimization, showing that representations learned by members of the identifiable model family approach equality up to a linear transformation as a function of dataset size, neural network capacity, and optimization progress.

## 2 MODEL FAMILY AND DATA DISTRIBUTION

The learned embeddings of a DNN are a function not only of the parameters, but also the network architecture and size of dataset (viewed as a sample from the underlying data distribution). This renders any analysis in full generality challenging. To make such an analysis tractable, in this section, we begin by specifying a set of assumptions about the underlying data distribution and model family that must hold for the learned representations to be similar up to a linear transformation. These assumptions are, in fact, satisfied by a number of already published, highly performant models. We establish definitions in this section, and discuss these existing approaches in depth in Section 4.

**Data Distribution** We assume the existence of a generalized dataset in the form of an empirical distribution $p_{\mathcal{D}}(\mathbf{x}, \mathbf{y}, \mathbf{S})$ over random variables $\mathbf{x}$, $\mathbf{y}$ and $\mathbf{S}$ with the following properties:

- The random variable $\mathbf{x}$ is an input variable, typically high-dimensional, such as text or an image.
- The random variable $\mathbf{y}$ is a target variable whose value the model predicts. In case of object classification, this would be some semantically meaningful class label. However, in our model family, $\mathbf{y}$ may also be a high-dimensional context variable, such a text, image, or sentence fragment.
- $\mathbf{S}$ is a set containing the possible values of $\mathbf{y}$ given $\mathbf{x}$, so $p_{\mathcal{D}}(\mathbf{y}|\mathbf{x}, \mathbf{S}) > 0 \iff \mathbf{y} \in \mathbf{S}$.

Note that the set of labels $\mathbf{S}$ is not fixed, but a random variable. This allows supervised, contrastive, and self-supervised learning frameworks to be analyzed together: the meaning of $\mathbf{S}$ encodes the task. For supervised classification, $\mathbf{S}$ is deterministic and contains class labels. For self-supervised pretraining, $\mathbf{S}$ contains randomly-sampled high-dimensional variables such as image embeddings. For deep metric learning (Hoffer and Ailon, 2015; Sohn, 2016), the set $\mathbf{S}$ contains one positive and $k$ negative samples of the class to which $\mathbf{x}$ belongs.

**Canonical Discriminative Form** Given a data distribution as above, a generalized discriminative model family may be defined by its parameterization of the probability of a target variable $\mathbf{y}$ conditioned on an observed variable $\mathbf{x}$ and a set $\mathbf{S}$ that contains not only the true target label $\mathbf{y}$, but

also a collection of distractors $\mathbf{y}'$:

$$p_{\boldsymbol{\theta}}(\mathbf{y}|\mathbf{x}, \mathbf{S}) = \frac{\exp(\mathbf{f}_{\boldsymbol{\theta}}(\mathbf{x})^\top \mathbf{g}_{\boldsymbol{\theta}}(\mathbf{y}))}{\sum_{\mathbf{y}' \in \mathbf{S}} \exp(\mathbf{f}_{\boldsymbol{\theta}}(\mathbf{x})^\top \mathbf{g}_{\boldsymbol{\theta}}(\mathbf{y}'))}, \tag{1}$$

The codomain of the functions $\mathbf{f}_{\boldsymbol{\theta}}(\mathbf{x})$ and $\mathbf{g}_{\boldsymbol{\theta}}(\mathbf{y})$ is $\mathbb{R}^M$, and the domains vary according to modelling task. For notational convenience both are parameterized by $\boldsymbol{\theta} \in \Theta$, but $\mathbf{f}$ and $\mathbf{g}$ may use disjoint parts of $\boldsymbol{\theta}$, meaning that they do not necessarily share parameters.

With $\mathcal{F}$ and $\mathcal{G}$ we denote the function spaces of $\mathbf{f}_{\boldsymbol{\theta}}$ and $\mathbf{g}_{\boldsymbol{\theta}}$ respectively. Our primary domain of interest is when $\mathbf{f}_{\boldsymbol{\theta}}$ and $\mathbf{g}_{\boldsymbol{\theta}}$ are highly flexible function approximators, such as DNNs. This brings certain analytical challenges. In neural networks, different choices of parameters $\boldsymbol{\theta}$ can result in the same functions $\mathbf{f}_{\boldsymbol{\theta}}$ and $\mathbf{g}_{\boldsymbol{\theta}}$, hence the map $\Theta \rightarrow \mathcal{F} \times \mathcal{G}$ is many-to-one. In the context of representation learning, the function $\mathbf{f}_{\boldsymbol{\theta}}$ is typically viewed as a nonlinear feature extractor, e.g., the learned representation of the input data. While other choices meet the membership conditions for the family defined by the canonical form of Equation (1), in the remainder, we will focus on DNNs in the remainder. We next present a definition of identifiability suitable for DNNs, and prove that members of the above family satisfy it under additional assumptions.

## 3 MODEL IDENTIFIABILITY

In this section, we derive identifiability conditions for models in the family defined in Section 2.

### 3.1 IDENTIFIABILITY IN PARAMETER SPACE

Identifiability analysis answers the question of whether it is theoretically possible to learn the parameters of a statistical model exactly. Specifically, given some estimator $\boldsymbol{\theta}'$ for model parameters $\boldsymbol{\theta}^*$, identifiability is the property that, for any $\{\boldsymbol{\theta}', \boldsymbol{\theta}^*\} \subset \Theta$,

$$p_{\boldsymbol{\theta}'} = p_{\boldsymbol{\theta}^*} \implies \boldsymbol{\theta}' = \boldsymbol{\theta}^*. \tag{2}$$

Models that do not have this property are said to be non-identifiable. This happens when different values $\{\boldsymbol{\theta}', \boldsymbol{\theta}^*\} \subset \Theta$ can give rise to the same model distribution $p_{\boldsymbol{\theta}'}(\mathbf{y}|\mathbf{x}, \mathbf{S}) = p_{\boldsymbol{\theta}^*}(\mathbf{y}|\mathbf{x}, \mathbf{S})$. In such a case, observing an empirical distribution $p_{\boldsymbol{\theta}^*}(\mathbf{y}|\mathbf{x}, \mathbf{S})$, and fitting a model $p_{\boldsymbol{\theta}'}(\mathbf{y}|\mathbf{x}, \mathbf{S})$ to it perfectly does not guarantee that $\boldsymbol{\theta}' = \boldsymbol{\theta}^*$.

Neural networks exhibit various symmetries in parameter space such that there is almost always a many-to-one correspondence between a choice of $\boldsymbol{\theta}$ and resulting probability function $p_{\boldsymbol{\theta}}$. A simple example in neural networks is that one can swap the (incoming and outgoing) connections of two neurons in a hidden layer. This changes the value of the parameters, but does not change the network's function. Thus, when representation functions $\mathbf{f}_{\boldsymbol{\theta}}$ or $\mathbf{g}_{\boldsymbol{\theta}}$ are parameterized as DNNs, equation 2 is not satisfiable.

### 3.2 IDENTIFIABILITY IN FUNCTION SPACE

For reliable and efficient representation learning, we want learned representations $\mathbf{f}_{\boldsymbol{\theta}}$ from two identifiable models to be *sufficiently* similar for interchangeable use in downstream tasks. The most general property we wish to preserve among learned representations is their ability to discriminate among statistical patterns corresponding to categorical groupings. In the model family defined in Section 2, the data and context functions $\mathbf{f}_{\boldsymbol{\theta}}$ and $\mathbf{g}_{\boldsymbol{\theta}}$ parameterize $p_{\boldsymbol{\theta}}(\mathbf{y}|\mathbf{x}, \mathbf{S})$, the probability of label assignment, through a normalized inner product. This induces a hyperplane boundary, for discrimination, in a joint space of learned representations for data $\mathbf{x}$ and context $\mathbf{y}$. Therefore, in the following, we will derive identifiability conditions *up to a linear transformation*, using a notion of similarity in parameter space inspired by Hyvärinen et al. (2018).

**Definition 1.** *Let $\overset{L}{\sim}$ be a pairwise relation on $\Theta$ defined as:*

$$\boldsymbol{\theta}' \overset{L}{\sim} \boldsymbol{\theta}^* \iff \begin{array}{l} \mathbf{f}_{\boldsymbol{\theta}'}(\mathbf{x}) = \boldsymbol{A}\mathbf{f}_{\boldsymbol{\theta}^*}(\mathbf{x}) \\ \mathbf{g}_{\boldsymbol{\theta}'}(\mathbf{y}) = \boldsymbol{B}\mathbf{g}_{\boldsymbol{\theta}^*}(\mathbf{y}) \end{array} \tag{3}$$

*where $\boldsymbol{A}$ and $\boldsymbol{B}$ are invertible $M \times M$ matrices.* See Appendix B for proof that $\overset{L}{\sim}$ is an equivalence relation. In the remainder, we refer to identifiability up to the equivalence relation $\overset{L}{\sim}$ as $\overset{L}{\sim}$-*identifiable* or *linearly identifiable*.

### 3.3 LINEAR IDENTIFIABILITY OF LEARNED REPRESENTATIONS

We next present a simple derivation of the $\overset{L}{\sim}$-identifiability of members of the generalized discriminative family defined in Section 2. This result reveals sufficient conditions under which a discriminative probabilistic model $p_{\boldsymbol{\theta}}(\mathbf{y}|\mathbf{x}, \mathbf{S})$ has a useful property: the learned representations of the input $\mathbf{x}$ and target random variables $\mathbf{y}$ for any two pairs of parameters $(\boldsymbol{\theta}', \boldsymbol{\theta}^*)$ are related as $\boldsymbol{\theta}' \overset{L}{\sim} \boldsymbol{\theta}^*$, that is, $\mathbf{f}_{\boldsymbol{\theta}'}(\mathbf{x}) = \boldsymbol{A}\mathbf{f}_{\boldsymbol{\theta}^*}(\mathbf{x})$ and $\mathbf{g}_{\boldsymbol{\theta}'}(\mathbf{y}) = \boldsymbol{B}\mathbf{g}_{\boldsymbol{\theta}^*}(\mathbf{y})$.

We first review the notation for the proof, which is introduced in detail in Section 2. We then highlight an important requirement on the diversity of the data distribution, which must be satisfied for the proof statement to hold. We prove the result immediately after.

**Notation.** The target random variables $\mathbf{y}$, associated with input random variables $\mathbf{x}$, may be class labels (as in supervised classification), or they could be stochastically generated from datapoints $\mathbf{x}$ as, e.g., perturbed image patches (as in self-supervised learning). We account for this additional stochasticity as a set-valued random variable $\mathbf{S}$, containing all possible values of $\mathbf{y}$ conditioned on some $\mathbf{x}$. For brevity, we will use shorthands that drop the parameters $\boldsymbol{\theta}$: $p' := p_{\boldsymbol{\theta}'}, p^* := p_{\boldsymbol{\theta}^*}$, $\mathbf{f}^* := \mathbf{f}_{\boldsymbol{\theta}^*}, \mathbf{f}' := \mathbf{f}_{\boldsymbol{\theta}'}, \mathbf{g}' := \mathbf{g}_{\boldsymbol{\theta}'}$.

**Diversity condition.** We assume that for any $(\boldsymbol{\theta}', \boldsymbol{\theta}^*)$ for which it holds that $p' = p^*$, and for any given $\mathbf{x}$, by repeated sampling $\mathbf{S} \sim p_{\mathcal{D}}(\mathbf{S}|\mathbf{x})$ and picking $\mathbf{y}_A, \mathbf{y}_B \in \mathbf{S}$, we can construct a set of $M$ distinct tuples $\{(\mathbf{y}_A^{(i)}, \mathbf{y}_B^{(i)})\}_{i=1}^M$ such that the matrices $\mathbf{L}'$ and $\mathbf{L}^*$ are invertible, where $\mathbf{L}'$ consists of columns $(\mathbf{g}'(\mathbf{y}_A^{(i)}) - \mathbf{g}'(\mathbf{y}_B^{(i)}))$, and $\mathbf{L}^*$ consists of columns $\mathbf{g}^*(\mathbf{y}_A^{(i)}) - \mathbf{g}^*(\mathbf{y}_B^{(i)})$, $i \in \{1, \ldots, M\}$. See Section 3.4 for detailed discussion.

**Theorem 1.** Under the diversity condition, models in the family defined by Equation (1) are linearly identifiable. That is, for any $\boldsymbol{\theta}', \boldsymbol{\theta}^* \in \Theta$, and $\mathbf{f}^*, \mathbf{f}', \mathbf{g}^*, \mathbf{g}', p^*, p'$ defined as in Section 2,

$$p' = p^* \implies \boldsymbol{\theta}' \overset{L}{\sim} \boldsymbol{\theta}^*. \tag{4}$$

To establish the result, we proceed by directly constructing an invertible linear transformation that satisfies Definition 1. Consider $\mathbf{y}_A, \mathbf{y}_B \in \mathbf{S}$. The likelihood ratios for these points

$$\frac{p'(\mathbf{y}_A|\mathbf{x}, \mathbf{S})}{p'(\mathbf{y}_B|\mathbf{x}, \mathbf{S})} = \frac{p^*(\mathbf{y}_A|\mathbf{x}, \mathbf{S})}{p^*(\mathbf{y}_B|\mathbf{x}, \mathbf{S})} \tag{5}$$

are equal. Substituting our model definition from equation (1), we find:

$$\frac{\exp(\mathbf{f}'(\mathbf{x})^\top \mathbf{g}'(\mathbf{y}_A))}{\exp(\mathbf{f}'(\mathbf{x})^\top \mathbf{g}'(\mathbf{y}_B))} = \frac{\exp(\mathbf{f}^*(\mathbf{x})^\top \mathbf{g}^*(\mathbf{y}_A))}{\exp(\mathbf{f}^*(\mathbf{x})^\top \mathbf{g}^*(\mathbf{y}_B))}, \tag{6}$$

where the normalizing constants cancelled out on the left- and right-hand sides. Taking the logarithm, this simplifies to:

$$(\mathbf{g}'(\mathbf{y}_A) - \mathbf{g}'(\mathbf{y}_B))^\top \mathbf{f}'(\mathbf{x}) = (\mathbf{g}^*(\mathbf{y}_A) - \mathbf{g}^*(\mathbf{y}_B))^\top \mathbf{f}^*(\mathbf{x}). \tag{7}$$

Note that this equation is true for any triple $(\mathbf{x}, \mathbf{y}_A, \mathbf{y}_B)$ for which $p_{\mathcal{D}}(\mathbf{x}, \mathbf{y}_B, \mathbf{y}_B) > 0$.

We next collect $M$ distinct tuples $(\mathbf{y}_A^{(i)}, \mathbf{y}_B^{(i)})$ so that by repeating Equation (7) $M$ times and by the diversity condition noted above, the resulting difference vectors are linearly independent. We collect these vectors together as the columns of $(M \times M)$-dimensional matrices $\mathbf{L}'$ and $\mathbf{L}^*$, forming the following system of $M$ linear equations:

$$\mathbf{L}'^\top \mathbf{f}'(\mathbf{x}) = \mathbf{L}^{*\top} \mathbf{f}^*(\mathbf{x}).$$

Since $\mathbf{L}'$ and $\mathbf{L}^*$ are invertible, we rearrange:

$$\mathbf{f}'(\mathbf{x}) = (\mathbf{L}^* \mathbf{L}'^{-1})^\top \mathbf{f}^*(\mathbf{x}). \tag{8}$$

Hence, $\mathbf{f}'(\mathbf{x}) = \mathbf{A}\mathbf{f}^*(\mathbf{x})$ where $\mathbf{A} = (\mathbf{L}^* \mathbf{L}'^{-1})$. This completes the first half of the proof. See Appendix C for the second half of the proof, which is similar, and handles the function $\mathbf{g}$.

### 3.4 DISCUSSION: WHEN DOES THE DIVERSITY CONDITION HOLD?

Theorem 1 is a constructive proof of existence that exhibits invertible $(M \times M)$ matrices $\mathbf{L}'$ and $\mathbf{L}^*$. We require the diversity condition to hold in order to guarantee invertibility. Such a requirement is similar to the conditions in earlier work on nonlinear ICA such as (Hyvärinen et al., 2018), as discussed in Section 6. Informally, this means that there needs to be a sufficient number of possible values $\mathbf{y} \in \mathbf{S}$. In the case of supervised classification with $K$ classes, $\mathbf{S}$ is fixed and of size $K$. Then, we need $K \geq M + 1$ in order to generate $M$ difference vectors $\mathbf{g}_{\boldsymbol{\theta}}(\mathbf{y}^{(1)}) - \mathbf{g}_{\boldsymbol{\theta}}(\mathbf{y}^{(j)})$, $j = 2, \ldots, M + 1$. In case of self-supervised or deep metric learning, where $\mathbf{S}$ and $\mathbf{y}$ may be algorithmically generated from $\mathbf{x}$, this requirement is easy to satisfy, as there will typically be a diversity of values of $\mathbf{y}$. The same holds for language models with large vocabularies. However, for supervised classification with a small number of classes, this requirement on the size of $\mathbf{S}$ may be restrictive, as we discuss further in Section 4.

Note that by placing the diversity requirement on the number of classes $K$, we implicitly assumed that the context representation function $\mathbf{g}_{\boldsymbol{\theta}}$ has the following property: the $M$ difference vectors span the range of $\mathbf{g}_{\boldsymbol{\theta}}$. This is a mild assumption in the context of DNNs: for random initialization and iterative weight updates, this property follows from the stochasticity of the distribution used to initialize the network. Briefly, a set of $M + 1$ unique points $\mathbf{y}^{(j)}$ such that the $M$ vectors $\mathbf{g}_{\boldsymbol{\theta}}(\mathbf{y}^{(1)}) - \mathbf{g}_{\boldsymbol{\theta}}(\mathbf{y}^{(j)}), j = 2, \ldots, M + 1$ are not linearly independent has measure zero. For other choices of $\mathbf{g}_{\boldsymbol{\theta}}$, care must be taken to ensure this condition is satisfied.

What can be said when $\mathbf{L}'$ and $\mathbf{L}^*$ are ill-conditioned, that is, the ratio between maximum and minimum singular value $\frac{\sigma_{\max}(\mathbf{L})}{\sigma_{\min}(\mathbf{L})}$ (dropping superscripts when a statement apply to both) is large? In the context of a data representation matrix such as $\mathbf{L}$, this implies that there exists at least one column $\boldsymbol{\ell}_j$ of $\mathbf{L}$ and constants $\lambda_k$ for $k \neq j$ such that $\|\boldsymbol{\ell}_j - \sum_{k \neq j} \lambda_k \boldsymbol{\ell}_k\|_2 < \varepsilon$ for small $\varepsilon$. In other words, some column is nearly a linear combination of the others. This implies, in turn, that there exists some tuple $(\mathbf{y}^{(k)}, \mathbf{y}^{(i)})$ such that the resulting difference vector $\boldsymbol{\ell}_j = \mathbf{g}_{\boldsymbol{\theta}}(\mathbf{y}_A^{(k)}) - \mathbf{g}_{\boldsymbol{\theta}}(\mathbf{y}_B^{(i)})$ can nearly (in the sense above) be written as a linear combination of the other columns. Such near singularity is in this case a function of the choice of samples $\mathbf{y}$ that yield the difference vectors. The issue could be handled by resampling different data points until the condition number of the matrices is satisfactory. This amounts to strengthening the diversity condition. We leave more detailed analysis to future work, as the result will depend on the choice of architectures for $\mathbf{f}$ and $\mathbf{g}$.

## 4 EXAMPLES OF LINEARLY IDENTIFIABLE MODELS

The form of Equation (1) is already used as a general approach for a variety of machine learning problems. We present a non-exhaustive sample of such publications, chosen to exhibit the range of applications. Many of these approaches were state-of-the-art at the time of their release: Contrastive Predictive Coding (Hénaff et al., 2019), BERT (Devlin et al., 2018), GPT-2 and GPT-3 (Radford et al., 2018; 2019; Brown et al., 2020), XLNET (Yang et al., 2019), and the triplet loss for deep metric learning (Sohn, 2016). In this section, we discuss how to interpret the functional components of these frameworks with respect to the generalized data distribution of Section 2 and canonical parameterization of Equation (1). See Appendix D for reductions to the canonical form of Equation (1).

**Supervised Classification.** Although the scope of this paper is identifiable representation learning, under certain conditions, standard supervised classifiers can learn identifiable representations as well. In this case, the number of classes must be strictly greater than the feature dimension, as noted in Section 3.4. We simulate such a model in Section 5.1 to show evidence of its linear identifiability. We stress that representation learning as *pretraining* for classification is a way to ensure that the conditions on label diversity are met, rather than relying on the supervised classifier itself to generate identifiable representations. This paradigm is discussed in the next subsection.

Representations learned during supervised classification can be linearly identifiable under the following model specification. The input random variables $\mathbf{x}$ represent some data domain to be classified, such as images or word embeddings. The target variables $\mathbf{y}$ represent label assignments for $\mathbf{x}$, typically semantically meaningful. These are often encoded these as the standard basis vectors $\mathbf{e_y}$, a "one-hot encoding." The set $\mathbf{S}$ contains all $K$ possible values of $\mathbf{y}$. In this case, notice that $\mathbf{S}$ is

not stochastic: the empirical distribution $p_{\mathcal{D}}(\mathbf{S}|\mathbf{x})$ is modelled as a Dirac measure with all probability mass on the set $\mathbf{S} = \{0, \ldots, K-1\}$ (using integers, here, to represent distinct labels) . The representation function $\mathbf{f}_{\boldsymbol{\theta}}(\mathbf{x})$ of a classifier is often implemented as DNN that maps from the input layer to the layer just prior to the model logits. The context map $\mathbf{g}_{\boldsymbol{\theta}}(\mathbf{y})$ is given by the weights in the final, linear projection layer, which outputs unnormalized logits. Concretely, $\mathbf{g}_{\boldsymbol{\theta}}(\mathbf{y}) = \mathbf{W}\mathbf{e}_{\mathbf{y}}$, where $\mathbf{W} \in \mathbb{R}^{M \times M}$ is a learnable weight matrix. In order satisfy the diversity condition, the dimension $M$ of the number of classes $K$ must be strictly greater than the dimension of the learned representation $M$, that is, $|\mathbf{S}| \geq M+1$. Finally, the output of the final, linear projection layer is normalized through a Softmax function, yielding the parameterization of Equation (1).

**Self-Supervised Pretraining for Image Classification.** Self-supervised learning is a framework that first pretrains a DNN before deploying it on some other, related task. The pretraining task often takes the form of Equation (1) and meets the sufficient conditions to be linearly identifiable. A paradigmatic example is Contrastive Predictive Coding (CPC) (Oord et al., 2018). CPC is a general pretraining framework, but we focus for the sake of clarity on its use in image models here. CPC as applied to images involves: (1) preprocessing an image into augmented patches, (2) assigning labels according to which image the patch came from, and then (3) predicting the representations of the patches whether below, to the right, to the left, or above a certain level (Oord et al., 2018).

The context function of CPC, $\mathbf{g}_{\boldsymbol{\theta}}(\mathbf{y})$, encodes a particular position in the sequence of patches, and the representation function, $\mathbf{f}_{\boldsymbol{\theta}}(\mathbf{x})$, is an autoregressive function of the previous $k$ patches, according to some predefined patch ordering. Given some $\mathbf{x}$, the collection of all patches from the sequence, from a given minibatch of images, is the set $\mathbf{S} \sim p_{\mathcal{D}}(\mathbf{S}|\mathbf{x})$, where the randomness enters via the patch preprocessing algorithm. Since the preprocessing phase is part of the algorithm design, it is straightforward to make it sufficiently diverse (enough transformations of enough patches) so as to meet the requirements for the model to be linearly identifiable.

**Multi-task Pretraining for Natural Language Generation.** Autoregressive language models, such as (Mikolov et al., 2010; Dai and Le, 2015) and more recently GPT-2 and GPT-3 (Radford et al., 2018; 2019; Brown et al., 2020), are typically also instances of the model family of Equation 1. Data points $\mathbf{x}$ are the past tokens, $\mathbf{f}_{\boldsymbol{\theta}}(\mathbf{x})$ is a nonlinear representation of the past estimated by either an LSTM (Hochreiter and Schmidhuber, 1997) or an autoregressive Transformer model (Vaswani et al., 2017), $\mathbf{y}$ is the next token, and $\mathbf{w}_i = \mathbf{g}_{\boldsymbol{\theta}}(\mathbf{y} = i)$ is a learned representation of the next token, often implemented as a simple look-up table, as in supervised classification.

BERT (Devlin et al., 2018) is also a member of the linearly identifiable family. This model pretrains word embeddings through a denoising autoencoder-like (Vincent et al., 2008) architecture. For a given sequence of tokenized text, some fixed percentage of the symbols are extracted and set aside, and their original values set to a special null symbol, "corrupting" the original sequence. The pretraining task in BERT is to learn a continuous representation of the extracted symbols conditioned on the remainder of the text. A transformer (Vaswani et al., 2017) function approximator is used to map from the corrupted sequence into a continuous space. The transformer network is the $\mathbf{f}_{\boldsymbol{\theta}}(\mathbf{x})$ function of Equation 1. The context map $\mathbf{g}_{\boldsymbol{\theta}}(\mathbf{y})$ is a lookup map into the learned basis vector for each token.

## 5 EXPERIMENTS

The derivation in Section 3 shows that, for models in the general discriminative family defined in Section 2, the functions $\mathbf{f}_{\boldsymbol{\theta}}$ and $\mathbf{g}_{\boldsymbol{\theta}}$ are identifiable up to a linear transformation given unbounded data and assuming model convergence. The question remains as to how close a model trained on finite data and without convergence guarantees will approach this limit. One subtle issue is that poor architecture choices (such as too few hidden units, or inadequate inductive priors) or insufficient data samples when training can interfere with model estimation and thereby linear identifiability of the learned representations, due to underfitting. In this section, we study this issue over a range of models, from low-dimensional language embedding and supervised classification (Figures 1 and 2 respectively) to GPT-2 (Radford et al., 2019), an approximately $1.5 * 10^9$-parameter generative model of natural language (Figure 4). See Appendix A and the code release for details needed to reproduce.

Through these experiments, we show that (1) in the small dimensional, large data regime, linearly identifiable models yield learned representations that lie approximately within a linear transformation

of each other (Figures 1 and 2) as predicted by Theorem 1; and (2) in the high dimensional, large data regime, linearly identifiable models yield learned representations that exhibit a strong trend towards linear identifiability. The learned representations approach a linear transformation of each other monotonically, as a function of dataset sample size, neural network capacity (number of hidden units), and optimization progress. In the case of GPT-2, which has benefited from substantial tuning by engineers to improve model estimation, we find strong evidence of linear identifiability.

**Measuring linear similarity between learned representations.** How can we measure whether pairs of learned representations live within a linear transformation of each other in function space? We adapt Canonical Correlation Analysis (CCA) (Hotelling, 1936) for this purpose, which finds the optimal linear transformations to maximize correlation among two random vectors. On a randomly selected held-out subset $\mathcal{B} \subset \mathcal{D}$ of the training data we compute $\mathbf{f}_{\boldsymbol{\theta}_1}(\mathcal{B})$ and $\mathbf{f}_{\boldsymbol{\theta}_2}(\mathcal{B})$ for two models with parameters $\boldsymbol{\theta}_1$ and $\boldsymbol{\theta}_2$ respectively. Assume without loss of generality that $\mathbf{f}_{\boldsymbol{\theta}_1}(\mathcal{B})$ and $\mathbf{f}_{\boldsymbol{\theta}_2}(\mathcal{B})$ are centered. CCA finds the optimal linear transformations $\boldsymbol{C}$ and $\boldsymbol{D}$ such that the pairwise correlations $\rho_i$ between the $i^{th}$ columns of $\boldsymbol{C}^\top \mathbf{f}_{\boldsymbol{\theta}_1}(\mathcal{B})$ and $\boldsymbol{D}^\top \mathbf{f}_{\boldsymbol{\theta}_2}(\mathcal{B})$ are maximized. We collect correlations together in $\boldsymbol{\rho}$. If after linear transformation the two matrices are aligned, the mean of $\boldsymbol{\rho}$ will be 1; if they are instead uncorrelated, then the mean of $\boldsymbol{\rho}$ will be 0. We use the mean of $\boldsymbol{\rho}$ as a proxy for the existence of a linear transformation between $\mathbf{f}_{\boldsymbol{\theta}_1}(\mathcal{B})$ and $\mathbf{f}_{\boldsymbol{\theta}_2}(\mathcal{B})$. For DNNs, it is a well known phenomenon that most of the variability in a learned representation tends to concentrate in a low-dimensional subspace, leaving many noisy, random dimensions (Morcos et al., 2018). Such random noise can result in spurious high correlations in CCA. A solution to this problem is to apply Principal Components Analysis (PCA) (Pearson, 1901) to each of the two matrices $\mathbf{f}_{\boldsymbol{\theta}_2}(\mathcal{B})$ and $\mathbf{f}_{\boldsymbol{\theta}_1}(\mathcal{B})$, projecting onto their top-$k$ principal components, before applying CCA. This technique is known as SVCCA (Raghu et al., 2017).

## 5.1 SIMULATION STUDY: CLASSIFICATION BY DNNS

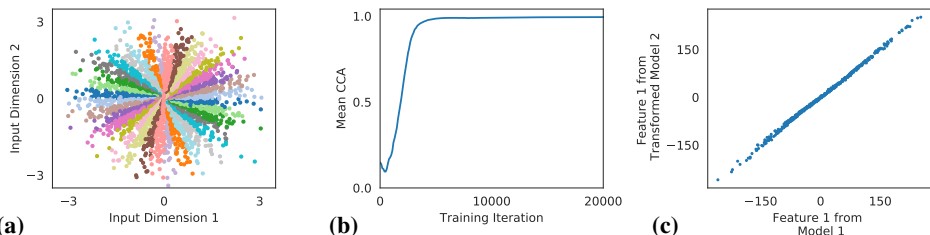

(a)  (b)  (c)

Figure 2: **Deep Supervised Classification**. (a) Data distribution for a linearly identifiable K-way classification problem. (b) Mean (centered) CCA between the learned representations over the course of training. After approx. 4000 iterations, CCA finds a linear transformation that rotate the learned representations into alignment, up to optimization error. (c) Learned representations after transformation via optimal linear transformation. The first dimension of the first model's feature space is plotted against the first dimension of second. The learned representations have a nearly linear relationship, modulo estimation noise.

We report first on a simulation study of linearly identifiable $K$-way classification, where all assumptions and sufficient conditions of Theorem 1 are guaranteed to be met. We generated a synthetic data distribution with the properties required by Section 2, and chose DNNs that had sufficient capacity to learn a specified nonlinear relationship between inputs $\mathbf{x}$ and targets $\mathbf{y}$. In short, the data distribution $p_{\mathcal{D}}(\mathbf{x}, \mathbf{y}, \mathbf{S})$ consists of inputs $\mathbf{x}$ sampled from a 2-D Gaussian with $\sigma = 3$. The targets $\mathbf{y}$ were assigned among $K = 18$ classes according to their radial position (angle swept out by a ray fixed at the origin). The number of classes $K$ was chosen to ensure $K \geq \dim[\mathbf{f}_{\boldsymbol{\theta}}(\mathbf{x})] + 1$, the diversity condition. See Appendix D.1 for more details.

To evaluate linear similarity, we trained two randomly initialized models of $p_{\mathcal{D}}(\mathbf{y}|\mathbf{x}, \mathbf{S})$. Plots show $\mathbf{f}_{\boldsymbol{\theta}}(\mathbf{x})$, the data representation function, on random $\mathbf{x}$. Figure 2b shows that the mean CCA increases to its maximum value over training, demonstrating that the feature spaces converge to the same solution up to a linear transformation modulo model estimation noise. Similarly, Figure 2c shows that the learned representations exhibit a strongly linear relationship.

## 5.2 SELF-SUPERVISED LEARNING FOR IMAGE CLASSIFICATION

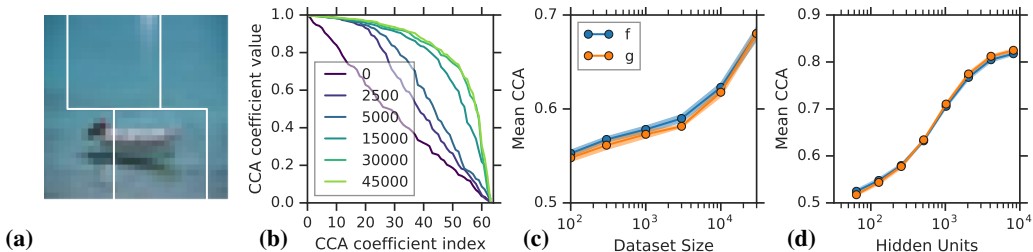

Figure 3: **Self-Supervised Representation Learning.** Error bars are computed over 5 pairs of models. **(a)** Input data. Two patches are taken (one from top half, and one from the bottom half) of an image at random. Using a contrastive loss, we predict the identity of the bottom patch encoding from the top. **(b)** Linear similarity of learned representations at checkpoints (see legend). As models converge, linear similarity increases. **(c)** Linear similarity as we increase the amount of data for $\mathbf{f}_\theta$ and $\mathbf{g}_\theta$. Error bars are computed over 5 pairs of models. **(d)** As we increase model size, linear similarity after convergence increases for both $\mathbf{f}_\theta$ and $\mathbf{g}_\theta$.

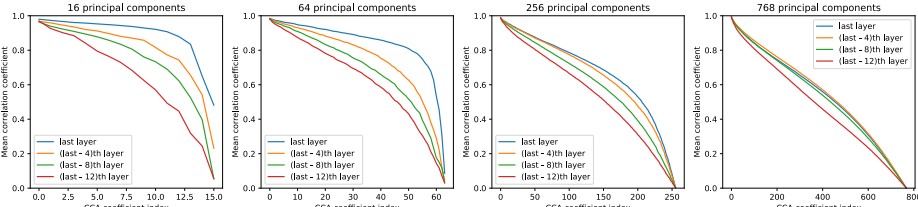

Figure 4: **Text Embeddings by GPT-2**. GPT-2 results. Representations of the last hidden layer (which is identifiable), in addition to three earlier layers (not necessarily identifiable) for four GPT-2 models. For each representation layer, SVCCA is computed over to all pairs of models, over which correlation coefficients were averaged. SVCCA was applied with 16, 64, 256 and 768 principal components. The learned representations in the *last*, identifiable layer more correlated than representations learned in preceding layers.

We next investigate high-dimensional, self-supervised representation learning on CIFAR-10 (Krizhevsky et al., 2009) using CPC (Oord et al., 2018; Hénaff et al., 2019). For a given input image, this model predicts the identity of a bottom image patch representation given a top patch representation (Figure 3a.) Here, **S** comprises the true patch with a set of distractor patches from across the current minibatch. For each model we define both $\mathbf{f}_{\theta'}$ and $\mathbf{g}_{\theta'}$ as a 3-layer MLP with 256 units per layer (except where noted otherwise) and fix output dimensionality of 64.

In Figure 3b, CCA coefficients are plotted over the course of training. As training progresses, alignment between the learned representations increases. In Figure 3c, we artificially limited the size of the dataset, and plot mean correlation after training and convergence. This shows that increasing availability of data correlates with closer alignment. In Figure 3d, we fix dataset size and artificially limit the model capacity (number hidden units) to investigate the effect of model size on the learned representations, varying the number of hidden units from 64 to 8192. This show that increasing model capacity correlates with increase in alignment of learned representations.

### 5.3 GPT-2

Finally, we report on a study of GPT-2 (Radford et al., 2019), a massive-scale language model. The identifiable representation is the set of features just before the last linear layer of the model. We use pretrained models from HuggingFace (Wolf et al., 2019). HuggingFace provides four different versions of the GPT-2: `gpt2`, `gpt2-medium`, `gpt2-large` and `gpt2-xl`, which differ mainly in the hyper-parameters that determine the width and depth of the neural network layers. For approximately 2000 input sentences, per timestep, for each model, we extracted representations at the last layer (which is identifiable) in addition to the representations per timestep given by three earlier layers in the model. Then, we performed SVCCA on each possible pair of models, on each of

the four representations. SVCCA was performed with 16, 64, 256 and 768 principal components, computed by applying SVD separately for each representations of each model. We chose 768 as the largest number of principal components, since that is the representation size for the smallest model in the repository (`gpt2`). We then averaged the CCA correlation coefficients across the pairs of models. Figure 4 shows the results. The results align well with our theory, namely that the representations at the last layer are more linearly related than the representations at other layers of the model.

### 5.4 INTERPRETATION AND SUMMARY

Theorem 1 establishes linear identifiability as an asymptotic property of a model that holds in the limit of infinite data and exact estimation. The experiments of this section have shown that for linear identifiable models, when the dimensionality is small relative to dataset size (Figures 1 and 2), the learned embeddings are closely linearly related, up to noise. Problems of model estimation and sufficient dataset size are more pronounced in high dimensions. Nevertheless, in GPT-2, representations among different trained models do in fact approach a mean correlation coefficient of 1.0 after training (Figure 4, blue line), providing strong evidence of linear identifiability.

## 6 RELATED WORKS

Prior to Hyvärinen and Morioka (2016), identifiability analysis was uncommon in deep learning. We build on advances in the theory of nonlinear ICA (Hyvärinen and Morioka, 2016; Hyvärinen et al., 2018; Khemakhem et al., 2019). In this section, we carefully distinguish our results from prior and concurrent works. Our diversity assumption is similar to diversity assumptions in these earlier works, while differing on certain conditions. The main difference is that their results apply to related but distinct families of models compared to the general discriminative family outlined in this paper. Arguably most related is Theorem 3 of Hyvärinen et al. (2018) and its proof, which shows that a class of contrastive discriminative models will estimate, up to an affine transformation, the true latent variables of a nonlinear ICA model. The main difference with our result is that they additionally assume: (1) that the mapping between observed variables and latent representations is invertible; and (2) that the discriminative model is binary logistic regression exhibiting universal approximation (Hornik et al., 1989), estimated with a contrastive objective. In addition, (Hyvärinen et al., 2018) does not present conditions for affine identifiability for their version of the context representation function $\mathbf{g}$. It should be noted that Theorem 1 in (Hyvärinen et al., 2018) provides a potential avenue for further generalization of our theorem 1 to discriminative models with non-linear interaction between $\mathbf{f}$ and $\mathbf{g}$.

Concurrent work (Khemakhem et al., 2020) has expanded the theory of identifiable nonlinear ICA to a class of conditional energy-based models (EBMs) with universal density approximation capability, therefore imposing milder assumptions than previous nonlinear ICA results. Their version of affine identifiability is similar to our result of linear identifiability in Section 3.2. The main differences are that Khemakhem et al. (2020) focus in both theory and experiment on EBMs. This allows for alternative versions of the diversity condition, assuming that the Jacobians of their versions of $\mathbf{f}$ or $\mathbf{g}$ are full rank. This is only possible if $\mathbf{x}$ or $\mathbf{y}$ are assumed continuous-valued; note that we do not make such an assumption. Khemakhem et al. (2020) also presents an architecture for which the conditions provably hold, in addition to sufficient conditions for identifiability up to element-wise scaling, which we did not explore in this work. While we build on these earlier results, we are, to the best of our knowledge, the first to apply identifiability analysis to state-of-the-art discriminative and autoregressive generative models.

## 7 CONCLUSION

We have shown that representations learned by a large family of discriminative models are identifiable up to a linear transformation, providing a novel perspective on representation learning using DNNs. Since identifiability is a property of a model class, and identification is realized in the asymptotic limit of data and compute, we perform experiments in the more realistic setting with finite datasets and finite compute. Our empirical results show that as the representational capacity of the model and dataset size increases, learned representations indeed tend towards solutions that are equal up to only a linear transformation.

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

## A    Reproducing Experiments and Figures

In this section, we present training and optimization details needed to reproduce our empirical validation of Theorem 1. We also published notebooks and check-pointed weights for two crucial experiments that investigate the result in the small and massive scale regimes, for Figure 1 and GPT-2 (ANONYMIZED).

### A.1    Figure 1

We provide a Jupyter notebook and model checkpoints for reproducing Figure 1. Please refer to this for hyperparameter settings. In short, we implemented a model (Mnih and Teh, 2012) in the family of Section 2 and trained it on the Billion Word dataset (Chelba et al., 2013). This is illustrative of the property of Theorem 1 because the relatively modest size of the parameter space (see notebook) and massive dataset minimizes model convergence and data availability restrictions, e.g., approaches the asymptotic regime.

The word embedding space is 2-D for ease of visualization. We randomly selected a subset of words, mapped them into their learned embeddings, and visualized them as points in the left and middle panes. We then regress pane one onto pane two in order to learn the best linear transformation between them. Note that if the two are linear transformations of each other, regression will recover that transformation exactly.

### A.2    Simulation Study: Classification by DNNs

For this experiment, we want to ensure that the chosen model can fit the data distribution exactly. Controlling this removes one possible factor that could prevent linear identifiability of learned representations despite the model formally having that property. We do this by making sure that the process that generates the dataset matches the model chosen to learn the relationships between inputs and labels.

This is achieved through the following algorithm. We first randomly assign initialization labels based on angular position, then fit two neural networks $f_{\theta^\star}$ and $g_{\theta^\star}$ to predict the final labels, using the discriminative model of Equation (1) and Appendix D.1. Both $f_{\theta^\star}$ and $g_{\theta^\star}$ 4-hidden-layer MLPs with two 64 unit layers and one 2-D bottle neck layer. After training these representation functions to convergence, generated new batch of points $\mathbf{x}$, and used the trained networks to predict the ground truth labels $\mathbf{y}$.

Finally, to conduct experiments, we chose $\mathbf{f}_{\theta'}$ and $\mathbf{g}_{\theta'}$ to be the same architecture as $\mathbf{f}_{\theta^\star}$ and $\mathbf{g}_{\theta^\star}$. This ensures that the supervised classifier we attempted to learn would using the function approximators $\mathbf{f}_{\theta'}$ and $\mathbf{g}_{\theta'}$ would be able to capture the true data generating process, e.g, would not fail due to too few hidden units, or too complex a relationship between targets and inputs.

Remaining training details are as follows. We optimize weights using Adam with a learning rate of $10^{-4}$ for $5 * 10^4$ iterations. To make the classification problem more challenging, we additionally add 20 input dimensions of random noise to the data. The Adam optimizer Kingma and Ba (2014) with a learning rate of $3 \cdot 10^{-4}$ is used.

### A.3    Self-Supervised Learning for Image Classification

To compute linear similarity between representations, we train two independent models in parallel. For each model we define both $\mathbf{f}_\theta$ and $\mathbf{g}_\theta$ as a 3-layer fully connected neural network with $2^8$ units per layer and a fixed output dimensionality of $2^6$. We define our model following Equation (1), where $S$ is the set of the other image patches from the current minibatch and optimize the objective of (Hénaff et al., 2019). We augment both sampled patches independently with randomized brightness, saturation, hue, and contrast adjustments, following the recipe of (Hénaff et al., 2019). We train on the CIFAR10 dataset (Krizhevsky et al., 2009) with batchsize $2^8$, using the Adam optimizer with a learning rate of $10^{-4}$ and the JAX (Bradbury et al., 2018) software package. For each model, we early stop based on a validation loss failing to improve further.

Additional details about the experiments that generated Figure 3:

**Figure 3 a.**  Patches are sampled randomly from training images.

**Figure 3 b.**  For each model, we train for at most $3 * 10^4$ iterations, early stopping when necessary based on validation loss.

**Figure 3 c.**  For each model, we train for at most $3 * 10^4$ iterations, early stopping when necessary based on validation loss.

**Figure 3 d.**  Error bars show standard error computed over 5 pairs of models after $1.5 * 10^4$ training iterations.

## A.4  GPT-2

We include all details through a notebook in the code release. Pretrained GPT-2 weights as specified in the main text are publicly available from HuggingFace Wolf et al. (2019).

## A.5  REMARK ON EFFECT OF INITIALIZATION AND HYPERPARAMETERS OF MODELS

One question that may be of interest is whether initialization affects whether learned representations will be within a linear transformation of each other. This depends on whether the optimization routines (like Adam, AdaGrad, etc.) are robust to wider initialization within a certain range. If so, model convergence will be unaffected. However, this cannot make up for poor initialization or poor optimization: just as in any deep neural network, a poor initialization and inadequate optimizer will interfere with learning the model parameters. In the case of a linearly identifiable model, means that the learned representations would not live within a linear transformation of each other (up to noise from model fitting), since the models have failed to converge to a reasonable solution for the task at hand.

When the hyperparameters of a DNN are changed, this changes the class of functions that the network can represent (i.e., the size and stride of convolution filters will change which input pixels could be correlated in deeper layers). Typically, hyperparameters are carefully tuned using cross validation based on held-out data. We did so in our experiments also. We expect that such a tuning procedure would yield hyperparameters that are as good as possible for the model to be optimized, allowing sufficient optimization so that the linear identifiability of the learned representations is realized. If the hyperparameters are sufficiently bad and optimization suffers, this will interfere with model fitting, and with linear identifiability of the learned representations also.

## B  PROOF THAT LINEAR SIMILARITY IS AN EQUIVALENCE RELATION

We claim that $\overset{L}{\sim}$ is an equivalence relation. It suffices to show that it is reflexive, transitive, and symmetric.

*Proof.* Consider some function $\mathbf{g}_{\boldsymbol{\theta}}$ and some $\boldsymbol{\theta}', \boldsymbol{\theta}^\star, \boldsymbol{\theta}^\dagger \subset \Theta$. Suppose $\boldsymbol{\theta}' \overset{L}{\sim} \boldsymbol{\theta}^\star$. Then, there exists an invertible matrix $\mathbf{B}$ such that $\mathbf{g}_{\boldsymbol{\theta}'}(\mathbf{x}) = \mathbf{B}\mathbf{g}_{\boldsymbol{\theta}^\star}(\mathbf{x})$. Since $\mathbf{g}_{\boldsymbol{\theta}^\star}(\mathbf{x}) = \mathbf{B}^{-1}\mathbf{g}_{\boldsymbol{\theta}'}(\mathbf{x})$, $\overset{L}{\sim}$ is symmetric. Reflexivity follows from setting $\mathbf{g}_{\boldsymbol{\theta}^\star}$ to $\mathbf{g}_{\boldsymbol{\theta}'}$ and $\mathbf{B}$ to the identity matrix. To show transitivity, suppose also that $\boldsymbol{\theta}^\star \overset{L}{\sim} \boldsymbol{\theta}^\dagger$. Then, there exists an invertible $\mathbf{C}$ such that $\mathbf{g}_{\boldsymbol{\theta}^\star}(\mathbf{x}) = \mathbf{C}\mathbf{g}_{\boldsymbol{\theta}^\dagger}(\mathbf{x})$. Since $\mathbf{g}_{\boldsymbol{\theta}'} \overset{L}{\sim} \mathbf{g}_{\boldsymbol{\theta}^\star}$, $\mathbf{B}^{-1}\mathbf{g}_{\boldsymbol{\theta}'}(\mathbf{x}) = \mathbf{C}\mathbf{g}_{\boldsymbol{\theta}^\dagger}(\mathbf{x})$. Rearranging terms, $\mathbf{g}_{\boldsymbol{\theta}'}(\mathbf{x}) = \mathbf{B}\mathbf{C}\mathbf{g}_{\boldsymbol{\theta}^\dagger}(\mathbf{x})$, so that $\boldsymbol{\theta}' \overset{L}{\sim} \boldsymbol{\theta}^\dagger$ as required. $\square$

## C  SECTION 3.2 CONTINUED: CASE OF CONTEXT REPRESENTATION FUNCTION g

Our derivation of identifiability of $\mathbf{g}_{\boldsymbol{\theta}}$ is similar to the derivation of $\mathbf{f}_{\boldsymbol{\theta}}$. The primary difference is that the normalizing constants in Equation (6) do not cancel out. First, note that we can rewrite Equation 1 as:

$$p_{\boldsymbol{\theta}}(\mathbf{y}|\mathbf{x}, \mathbf{S}) = \exp(\widetilde{\mathbf{f}}_{\boldsymbol{\theta}}(\mathbf{x}, \mathbf{S})^\top \widetilde{\mathbf{g}}_{\boldsymbol{\theta}}(\mathbf{y})) \tag{9}$$

where:

$$\widetilde{\mathbf{f}}_{\boldsymbol{\theta}}(\mathbf{x}, \mathbf{S}) = [-Z(\mathbf{x}, \mathbf{S}); \mathbf{f}_{\boldsymbol{\theta}}(\mathbf{x})] \tag{10}$$

$$\widetilde{\mathbf{g}}_{\boldsymbol{\theta}}(\mathbf{y}) = [1; \mathbf{g}_{\boldsymbol{\theta}}(\mathbf{y})] \tag{11}$$

$$Z(\mathbf{x}, \mathbf{S}) = \log \sum_{\mathbf{y}' \in \mathbf{S}} \exp(\mathbf{f}_{\boldsymbol{\theta}}(\mathbf{x})^{\top} \mathbf{g}_{\boldsymbol{\theta}}(\mathbf{y}')). \tag{12}$$

Below, we will show that for the model family defined in Section 2,

$$p_{\boldsymbol{\theta}'} = p_{\boldsymbol{\theta}^*} \implies \mathbf{g}_{\boldsymbol{\theta}'}(\mathbf{y}) = \mathbf{B}\, \mathbf{g}_{\boldsymbol{\theta}^*}(\mathbf{y}), \tag{13}$$

where $\mathbf{B}$ is an invertible $(M \times M)$-dimensional matrix, concluding the proof of the linear identifiability of models in the family defined by Equation (1). We adopt the same shorthands as in the main text.

## C.1 DIVERSITY CONDITION

We assume that for any $(\boldsymbol{\theta}', \boldsymbol{\theta}^*) \subset \Theta$ for which it holds that $p' = p^*$, and for any given $\mathbf{y}$, there exist $M+1$ tuples $\{(\mathbf{x}^{(i)}, \mathbf{S}^{(i)})\}_{i=0}^{M}$, such that $p_{\mathcal{D}}(\mathbf{x}^{(i)}, \mathbf{y}, \mathbf{S}^{(i)}) > 0$, and such that the $((M+1) \times (M+1))$ matrices $\mathbf{M}'$ and $\mathbf{M}^*$ are invertible, where $\mathbf{M}'$ consists of columns $\widetilde{\mathbf{f}}'(\mathbf{x}^{(i)}, \mathbf{S}^{(i)})$, and $\mathbf{M}^*$ consists of columns $\widetilde{\mathbf{f}}^*(\mathbf{x}^{(i)}, \mathbf{S}^{(i)})$.

This is similar to the diversity condition of Section 3.2 but milder, since a typical dataset will have multiple $\mathbf{x}$ for each $\mathbf{y}$.

## C.2 PROOF

With the data distribution $p_{\mathcal{D}}(\mathbf{x}, \mathbf{y}, \mathbf{S})$, for a given $\mathbf{y}$, there exists a conditional distribution $p_{\mathcal{D}}(\mathbf{x}, \mathbf{S}|\mathbf{y})$. Let $(\mathbf{x}, \mathbf{S})$ be a sample from this distribution. From equation 1 and the statement to prove, it follows that:

$$p'(\mathbf{y}|\mathbf{x}, \mathbf{S}) = p^*(\mathbf{y}|\mathbf{x}, \mathbf{S}) \tag{14}$$

Substituting in the definition of our model from equation (9), we find:

$$\exp(\widetilde{\mathbf{f}}'(\mathbf{x}, \mathbf{S})^{\top} \widetilde{\mathbf{g}}'(\mathbf{y})) = \exp(\widetilde{\mathbf{f}}^*(\mathbf{x}, \mathbf{S})^{\top} \widetilde{\mathbf{g}}^*(\mathbf{y})), \tag{15}$$

which, evaluating logarithms, becomes

$$\widetilde{\mathbf{f}}'(\mathbf{x}, \mathbf{S})^{\top} \widetilde{\mathbf{g}}'(\mathbf{y}) = \widetilde{\mathbf{f}}^*(\mathbf{x}, \mathbf{S})^{\top} \widetilde{\mathbf{g}}^*(\mathbf{y}), \tag{16}$$

which is true for any triple $(\mathbf{x}, \mathbf{y}, \mathbf{S})$ where $p_{\mathcal{D}}(\mathbf{y}|\mathbf{x}, \mathbf{S}) > 0$.

From $\mathbf{M}'$ and $\mathbf{M}^*$ (Section C.1) and equation 16 we form a linear system of equations, collecting the $M + 1$ relationships together:

$$\mathbf{M}'^{\top} \widetilde{\mathbf{g}}'(\mathbf{y}) = \mathbf{M}^{*\top} \widetilde{\mathbf{g}}^*(\mathbf{y}) \tag{17}$$

$$\widetilde{\mathbf{g}}'(\mathbf{y}) = \mathbf{A} \widetilde{\mathbf{g}}^*(\mathbf{y}), \tag{18}$$

where $\mathbf{A} = (\mathbf{M}^* \mathbf{M}'^{-1})^{\top}$, an invertible $(M + 1) \times (M + 1)$ matrix.

It remains to show the existence of an invertible $M \times M$ matrix $\mathbf{B}$ such that

$$\mathbf{g}'(\mathbf{y}) = \mathbf{B}\mathbf{g}^*(\mathbf{y}). \tag{19}$$

We proceed by constructing $\mathbf{B}$ from $\mathbf{A}$. Since $\mathbf{A}$ is invertible, there exist $j$ elementary matrices $\{\mathbf{E}_1, \ldots, \mathbf{E}_j\}$ such that their action $\mathbf{R} = \mathbf{E}_j \mathbf{E}_{j-1} \ldots \mathbf{E}_1$ converts $\mathbf{A}$ to a (non-unique) row echelon form. Without loss of generality, we build $\mathbf{R}$ such that the $a_{1,1}$ entry of $\mathbf{A}$ is the first pivot, leading to the particular row echelon form:

$$\mathbf{RA} = \begin{bmatrix} a_{1,1} & a_{1,2} & a_{1,3} & \ldots & a_{1,m \times 1} \\ 0 & \tilde{a}_{2,2} & \tilde{a}_{2,3} & \ldots & \tilde{a}_{2,m \times 1} \\ 0 & 0 & \tilde{a}_{3,3} & \ldots & \tilde{a}_{2,m \times 1} \\ \vdots & \vdots & \vdots & \ddots & \vdots \\ 0 & 0 & \ldots & 0 & \tilde{a}_{m \times 1, m \times 1} \end{bmatrix}, \tag{20}$$

where $\tilde{a}_{i,j}$ indicates that the corresponding entry in $\mathbf{RA}$ may differ from $\mathbf{A}$ due to the action of $\mathbf{R}$. Applying $\mathbf{R}$ to Equation (17), we have

$$\mathbf{R}\widetilde{\mathbf{g}}'(\mathbf{y}) = \mathbf{RA}\widetilde{\mathbf{g}}^*(\mathbf{y}). \tag{21}$$

We now show that removing the first row and column of $\mathbf{RA}$ and $\mathbf{R}$ generates matrices of rank $M$. Let $\overline{\mathbf{RA}}$ and $\overline{\mathbf{R}}$ denote the $(M \times M)$ submatrices formed by removing the first row and column of $\mathbf{RA}$ and $\mathbf{R}$ respectively.

Equation (20) shows that $\overline{\mathbf{RA}}$ has a pivot in each column, and thus has rank $M$. To show that $\overline{\mathbf{R}}$ is invertible, we must show that removing the first row and column reduces the rank of $\mathbf{R} = \mathbf{E}_j \mathbf{E}_{j-1} \ldots \mathbf{E}_1$ by exactly 1. Clearly, each $\mathbf{E}_k$ is invertible, and their composition is invertible. We must show the same for the composition of $\overline{\mathbf{E}_k}$.

There are three cases to consider, corresponding to the three unique types of elementary matrices. Each elementary matrix acts on $\mathbf{A}$ by either (1) swapping rows $i$ and $j$, (2) replacing row $j$ by a multiple $m$ of itself, or (3) adding a multiple $m$ of row $i$ to row $j$. We denote elementary matrix types by superscripts.

In Case (1), $\mathbf{E}_k^1$ is an identity matrix with row $i$ and row $j$ swapped. For Case (2), $\mathbf{E}_l^2$ is an identity matrix with the $j, j^{th}$ entry replaced by some $m$. For each $\mathbf{E}_k^1$ and $\mathbf{E}_l^2$ in $\mathbf{R}$ , where $1 \le k, l \le j$, we know that the indices $i, j \ge 2$, because we chose the first entry of the first row of $\mathbf{A}$ to be the pivot, and hence do not swap the first row, or replace the first row by itself multiplied by a constant. This implies that removing the first row and column of $\mathbf{E}_k^1$ and $\mathbf{E}_l^2$ removes a pivot entry 1 in the $(1,1)$ position, and removes zeros elsewhere. Hence, the $(M \times M)$ submatrices $\overline{\mathbf{E}_k^1}$ and $\overline{\mathbf{E}_l^2}$ are elementary matrices with rank $M$.

For Case (3), $\mathbf{E}_k^3$ has some value $m \in \mathbb{R}$ in the $j, i^{th}$ entry, and 1s along the diagonal. In this case, we may find a non-zero entry in some $\mathbf{E}_k^3$, so that, e.g., the second row has a pivot at position $(2,2)$. Without loss of generality, suppose $i = 1$, $j = 2$ and let $m$ be some nonzero constant. Removing the first row and column of $\mathbf{E}_1^3$ removes this $m$ also. Nevertheless, $\overline{\mathbf{E}_1^3} = \mathbf{I}_M$, the rank $M$ identity matrix. For any other $\mathbf{E}_k^3$ $1 < i \le M + 1$, $j \ge 2$ because we chose $a_{1,1}$ as the first pivot, and hence do not swap the first row, or replace the first row by itself multiplied by a constant. In both cases, removing the first row and first column creates an $\overline{\mathbf{E}_k^3}$ that is a rank $M$ elementary matrix.

We have shown by the above that $\overline{\mathbf{R}}$ is a composition of rank $M$ matrices. We now construct the matrix $\mathbf{B}$ by removing the first entries of $\widetilde{\mathbf{g}}'$ and $\widetilde{\mathbf{g}}^\star$, and removing the first row and first column of $\mathbf{R}$ and $\mathbf{RA}$ in Equation (equation 21). Then, we have

$$\overline{\mathbf{R}}\mathbf{g}'(\mathbf{y}) = \overline{\mathbf{RA}}\mathbf{g}^*(\mathbf{y}), \tag{22}$$

$$\mathbf{g}'(\mathbf{y}) = \overline{\mathbf{R}}^{-1}\overline{\mathbf{RA}}\mathbf{g}^*(\mathbf{y}). \tag{23}$$

Choosing $\mathbf{B} = \overline{\mathbf{R}}^{-1}\overline{\mathbf{RA}}$ proves the result.

$\square$

## D  REDUCTIONS TO CANONICAL FORM OF EQUATION (1)

In the following, we show membership in the model family of Equation 1 using the mathematical notation of the papers under discussion in Section 4. Note that each subsection will change notation to match the papers under discussion, which varies quite widely. We employ the following colour-coding scheme to aid in clarity:

$$\log p_{\boldsymbol{\theta}}(\mathbf{y}|\mathbf{x}, \mathbf{S}) = \mathbf{f}_{\boldsymbol{\theta}}(\mathbf{x})^\top \mathbf{g}_{\boldsymbol{\theta}}(\mathbf{y}) - \log \sum_{\mathbf{y}' \in \mathbf{S}} \exp(\mathbf{f}_{\boldsymbol{\theta}}(\mathbf{x})^\top \mathbf{g}_{\boldsymbol{\theta}}(\mathbf{y}')),$$

where $\mathbf{f}_{\boldsymbol{\theta}}(\mathbf{x})$ is generalized to a data representation function, $\mathbf{g}_{\boldsymbol{\theta}}(\mathbf{y})$ is generalized to a context representation function, and $\sum_{\mathbf{y}' \in \mathbf{S}} \exp(\mathbf{f}_{\boldsymbol{\theta}}(\mathbf{x})^\top \mathbf{g}_{\boldsymbol{\theta}}(\mathbf{y}'))$ is some constant.

### D.1 SUPERVISED CLASSIFICATION

Supervised classifiers commonly employ a neural network feature extractor followed by a linear projection of the output of this network into a space of unnormalized logits. All the layers prior to the logits are the representation function $\mathbf{f_\theta}$, and the final projection layer is the context map $\mathbf{g_\theta}(y = i) = \mathbf{w}_i$, where $\mathbf{w}_i$ is the $i$-th column of a weight matrix $\mathbf{W}$. The set $\mathbf{S}$ in this case contains human-chosen labels and has no stochasticity. The loss function is the negative log-likelihood of the data under a categorical distribution with a softmax parameterization:

$$\log p_{\boldsymbol{\theta}}(y = i|\mathbf{x}; \mathbf{S}) = \mathbf{f_\theta}(\mathbf{x})^\top \boldsymbol{w}_i - \log \sum_{j=1}^{|\mathbf{S}|} \exp(\mathbf{f_\theta}(\mathbf{x})^\top \boldsymbol{w}_j)$$

Supervised classification is thus an member of the family defined in Section 2. It exhibits the simplest functional form for the $\mathbf{g}$ function while allowing $\mathbf{f}$ to be arbitrarily complicated.

### D.2 CPC

Consider a sequence of points $\mathbf{x}_t$. We wish to learn the parameters $\phi$ to maximize the $k$-step ahead predictive distribution $p(\mathbf{x}_{t+k}|\mathbf{x}_t, \phi)$. In the image patch example, each patch center $i, j$ is indexed by $t$. Each $\mathbf{x}_t$ is mapped to a sequence of feature vectors $\mathbf{z}_t = f_\theta(\mathbf{x}_t)$ An autoregressive model, already updated with the previous latent representations $\mathbf{z}_{\leq t-1}$, transforms the $\mathbf{z}_t$ into a "context" latent representation $\mathbf{c}_t = g_{AR}(\mathbf{z}_{\leq t})$. Instead of predicting future observations $k$ steps ahead, $\mathbf{x}_{t+k}$, directly through a generative model $p_k(\mathbf{x}_{t+k}|\mathbf{c}_t)$, Oord et al. (2018) model a density ratio in order to preserve the mutual information between $\mathbf{x}_{t+k}$ and $\mathbf{c}_t$.

**Objective** Let $\mathbf{X} = \{\mathbf{x}_1, \dots, \mathbf{x}_N\}$ be a set of $N$ random samples containing one positive sample from $p(\mathbf{x}_{t+k}|\mathbf{c}_t)$ and $N-1$ samples from the proposal distribution $p(\mathbf{x}_{t+k})$. Oord et al. (2018) define the following link function: $l_k(\mathbf{x}_{t+k}, \mathbf{c}_t) \triangleq \exp\left(\mathbf{z}_{t+k}^\top \mathbf{W}_k \mathbf{c}_t\right)$. Then, CPC optimizes

$$-\mathbb{E}_{\mathbf{X}} \left[\log \frac{l_k(\mathbf{x}_{t+k}, \mathbf{c}_t)}{\sum_{x_j \in X} l_k(\mathbf{x}_j, \mathbf{c}_t)}\right] = -\mathbb{E}_{\mathbf{X}} \left[\log \frac{\exp\left(\mathbf{z}_{t+k}^\top \mathbf{W}_k \mathbf{c}_t\right)}{\sum_{\mathbf{x}_j \in \mathbf{X}} \exp\left(\mathbf{z}_j^\top \mathbf{W}_k \mathbf{c}_t\right)}\right]. \quad (24)$$

Substituting in the definition of $l_k$ makes equation (24) identical to the model family (Equation 1).

### D.3 AUTOREGRESSIVE LANGUAGE MODELS (E.G. GPT-2)

Let $\mathcal{U} = \{u_1, \dots, u_n\}$ be a corpus of tokens. Autoregressive language models maximize a log-likelihood $L(\mathcal{U}) = \sum_{i=1}^{n} \log P(u_i|u_{i-k}, \dots, u_{i-1}; \Theta)$, Concretely, the conditional density is modelled as

$$\log P(u_i|u_{i-k:i-1}; \Theta)$$
$$= \mathbf{W}_{i:} \mathbf{h}_i - \log \sum_j \exp(\mathbf{W}_{j:} \mathbf{h}_i),$$

where $\mathbf{h}_i$ is the $m \times 1$ output of a function approximator (e.g. a Transformer decoder (Liu et al., 2018)), and $\mathbf{W}_{i:}$ is the $i$'th row of the $|\mathcal{U}| \times m$ token embedding matrix.

### D.4 BERT

Consider a sequence of text $\mathbf{x} = [x_1, \dots, x_T]$. Some proportion of the symbols in $\mathbf{x}$ are extracted into a vector $\bar{\mathbf{x}}$, and then set in $\mathbf{x}$ to a special null symbol, "corrupting" the original sequence. This operation generates the corrupted sequence $\underline{\mathbf{x}}$. The representational learning task is to predict $\bar{\mathbf{x}}$ conditioned on $\underline{\mathbf{x}}$, that is, to maximize w.r.t. $\boldsymbol{\theta}$:

$$\log p_\theta(\bar{\mathbf{x}}|\underline{\mathbf{x}}) \approx \sum_{t=1}^{T} m_t \log p_\theta(x_t|\underline{\mathbf{x}}) = \sum_{t=1}^{T} m_t \left( H_\theta(\underline{\mathbf{x}})_t^\top e(x_t) - \log \sum_{x'} \exp\left(H_\theta(\underline{\mathbf{x}})_t^\top e(x')\right)\right),$$

where $H$ is a transformer, $e$ is a lookup table, and $m_t = 1$ if symbol $x_t$ is masked. That is, corrupted symbols are "reconstructed" by the model, meaning that their index is predicted. As noted in Yang et al. (2019), BERT models the joint conditional probability $p(\bar{\mathbf{x}}|\mathbf{x})$ as factorized so that each masked token is separately reconstructed. This means that the log likelihood is approximate instead of exact.

### D.5 QUICKTHOUGHT VECTORS

Let $\mathbf{f}$ and $\mathbf{g}$ be functions that take a sentence as input and encode it into an fixed length vector. Let $s$ be a given sentence, and $S_{ctxt}$ be the set of sentences appearing in the context of $s$ for a fixed context size. Let $S_{cand}$ be the set of candidate sentences considered for a given context sentence $s_{ctxt} \in S_{ctxt}$. Then, $S_{cand}$ contains a valid context sentence $s_{ctxt}$ as well as many other non-context sentences. $S_{cand}$ is used for the classification objective. For any given sentence position in the context of $s$ (for example, the preceding sentence), the probability that a candidate sentence $s_{cand} \in S_{cand}$ is the correct sentence for that position is given by

$$\log p(s_{cand}|s, S_{cand}) = \boxed{f_\theta(s)^\top g_\theta(s_{cand}))} - \boxed{\log \sum_{s' \in S_{cand}} \exp\left(f_\theta(s)^\top g_\theta(s'_{cand})\right)}.$$

### D.6 DEEP METRIC LEARNING

The *multi-class N-pair loss* in Sohn (2016) is proportional to

$$\log N - \frac{1}{N} \sum_{i=1}^{N} \log\left(1 + \sum_{j \neq i} \exp\{\mathbf{f}_\theta(x_i)^\top \mathbf{f}_\theta(y_j) - \mathbf{f}_\theta(x_i)^\top \mathbf{f}_\theta(y_i))\}\right),$$

which can be simplified as

$$-\frac{1}{N} \sum_{i=1}^{N} \log\left(\frac{1}{K} \sum_{j=1}^{K} \exp\{\mathbf{f}_\theta(x_i)^\top \mathbf{f}_\theta(y_j) - \mathbf{f}_\theta(x_i)^\top \mathbf{f}_\theta(y_i)\}\right)$$

$$= \frac{1}{N} \sum_{i=1}^{N} \log\left(\frac{1}{\frac{1}{K} \sum_{j=1}^{K} \exp\{\mathbf{f}_\theta(x_i)^\top \mathbf{f}_\theta(y_j) - \mathbf{f}_\theta(x_i)^\top \mathbf{f}_\theta(y_i)\}}\right)$$

$$= \frac{1}{N} \sum_{i=1}^{N} \log\left(\frac{\exp\{\mathbf{f}_\theta(x_i)^\top \mathbf{f}_\theta(y_i)\}}{\frac{1}{K} \sum_{j=1}^{K} \exp\{\mathbf{f}_\theta(x_i)^\top \mathbf{f}_\theta(y_j)\}}\right).$$

Setting N to 1 and evaluating the log gives

$$\boxed{\mathbf{f}_\theta(x_i)^\top \mathbf{f}_\theta(y_i)} - \boxed{\frac{1}{K} \sum_{j=1}^{K} \exp(\mathbf{f}_\theta(x_i)^\top \mathbf{f}_\theta(y_j))},$$

which is Equation 1 where $\mathbf{f}_\theta = \mathbf{g}_\theta$.

### D.7 NEURAL PROBABILISTIC LANGUAGE MODELS (NPLMS)

Figure 1 shows results from a neural probabilistic language model as proposed in Mnih and Teh (2012). Mnih and Teh (2012) propose using a log-bilinear model (Mnih and Hinton, 2009) which, given some context $h$, learns a context word vectors $r_w$ and target word vectors $q_w$. Two different embedding matrices are maintained, in other words: one to capture the embedding of the word and the other the context. The representation for the context vector, $\hat{q}$, is then computed as the linear combination of the context words and a context weight matrix $C_i$ so that $\hat{q} = \sum_{i=1}^{n-1} C_i r_{w_i}$. The score for the match between the context and the next word is computed as a dot product, e.g., $s_\theta(w, h) = \hat{q}^\top \tilde{q}_w$[1] and substituting into the definition of $P_\theta^h(w)$, we see that

$$\log P_\theta^h(w) = \boxed{\hat{q}^\top \tilde{q}_w} - \boxed{\log \sum_{w'} \exp\left(\hat{q}^\top \tilde{q}_{w'}\right)}$$

---

[1]We have absorbed the per-token baseline offset $b$ into the $q_w$ defined in Mnih and Teh (2012), forming the vector $\tilde{q}_w$ whose $i$'th entry is $(q_w)_i = (q_w)_i + b_w/(\hat{q})_i$

shows that Mnih and Teh (2012) is a member of the model family.

Interestingly, a touchstone work in the area of NPLMs, Word2Vec (Mikolov et al., 2013), does not fall under the model family due to an additional nonlinearity applied to the score of Mnih and Teh (2012).

