# OpenReview forum: "On Linear Identifiability of Learned Representations"
_ICLR.cc/2021/Conference — Reject_

### Official Review · AnonReviewer3 · 2020-10-19
**Interesting experiments but some concerns on the theoretical claims**

**Rating:** 6
**Confidence:** 3

**Review:**

In this paper, the authors address the model identifiability in a general setting that can be adapted to several recent deep learning models (DNN supervised learning, CPC, BERT and GPT). Since model parameters (NN weights) are not identifiable, the authors hypothesize that vector f and g can be identifiably up to a linear transformation. Although the purpose of the work is appealing, there are some issues related to the current structure of the paper, proposed theory and its relationship to the provided experiments. See my detailed comments and questions below:
-	Figure 1: This figure is not referenced in the text. It should be referenced as a motivating fact.
-	Figure 1: It is mentioned in the Appendix that the subset of words was selected. How these subsets were chosen? Randomly? I think this is important to clarify.
-	Theoretical claims: This figure clearly illustrate a strong relationship between f’ and f*, which is close to a linear mapping, but it is not exactly linear. The main result of the paper is that if diversity condition is met, then models are linearly identifiable, but nothing is said about approximately linearly identifiable models as the ones shown in Figure 1. One question that should be addressed is how a perturbation in the diversity condition is translated to the identifiability property. It would be good that the authors add some theory about it.
-	It is suggested that eq. (1) holds for arbitrary supervised learning problem but, in the Appendix, the equivalence is not shown for this case. Also, the example provided in Fig. 2 seems to be a very arbitrary case. Is it possible to write any supervised learning problem as eq. (1)? Could you please include its derivation?
-	Is the form of eq. (1) already used as a general approach for different ML problems as the authors claim? If so, could you please cite relevant references?
-	Diversity condition means that such invertible matrices L’ and L* can be constructed from data samples. What can be said if those matrices are invertible but ill-conditioned, i.e. smallest singular value close to zero?
-	For supervised learning the condition implies that K>= M+1. Is it not a very restrictive condition? For example, it is possible to apply this analysis to a simple 2-classes supervised classification setting.
-	Agreement between theory and experimental results. With exception of the first experiment (supervised learning) the rest of the experiments show that by increasing iterations, dataset size and number of hidden units, then functions f and g tends to be linearly related (but not closely linear related). I don’t see why the main theoretical result of the paper (Theorem 1) is related to the experimental observations. I disagree with the authors claim that “these experiments validate Theorem 1”

---

> ### Author Response · Authors · 2020-11-17
> **Thank you for the careful reading and criticism; comments requested (2 of 2)**
>
> > Diversity condition means that such invertible matrices L’ and L* can be constructed from data samples. What can be said if those matrices are invertible but ill-conditioned, i.e. smallest singular value close to zero?
>
> Thanks for this interesting question! If the invertible matrices L’ and L* are ill-conditioned and the diversity condition holds, this means that at least one column could nearly be written as a linear combination of the others, in each matrix respectively.  This implies that there exists some tuple (y_A^(i), y_B^(i)) such that  the resulting difference vector d_i = g(y_A^(i)) - g(y_B^(i)) can almost be written as a linear combination of the other columns. Since the ill conditioning in this case is a function of the choice of points that yield the difference vectors (matrix columns), the problem could be handled by resampling different data points until the condition number of the matrices are satisfactory. This amounts to strengthening the diversity condition. We will add the above to the paper.
>
> > For supervised learning the condition implies that K>= M+1. Is it not a very restrictive condition? For example, it is possible to apply this analysis to a simple 2-classes supervised classification setting.
>
> Yes, we agree that this is a rather restrictive condition. Our paper is focused on representation learning using, e.g., self-supervised learning. In this case, the set S and possible values of the target variable y are very large: the elements in S are generated stochastically for each datapoint x in the batch (i.e., random patches of some window size from x, further processed with rotations and other noise-inducing transforms), making k >= M+1 easy to satisfy. This is also true for many language tasks, as the output vocabulary is much larger than the features dimension.
>
> >Agreement between theory and experimental results. With exception of the first experiment (supervised learning) the rest of the experiments show that by increasing iterations, dataset size and number of hidden units, then functions f and g tends to be linearly related (but not closely linear related). I don’t see why the main theoretical result of the paper (Theorem 1) is related to the experimental observations.
>
> We apologize for the lack of clarity about how the experimental results relate to Theorem 1. We attempted to address this in the first paragraph of the experiments section, we wrote:
>
> "The derivation in Section 3 shows that, for models in the general discriminative family defined in
> Section 2, the functions f and g are identifiable up to a linear transformation given unbounded data
> and assuming model convergence. The question remains as to how close a model trained on finite
> data and without convergence guarantees will approach this limit–that is, the domain of deep learning
> practice. Results in this section present evidence for: (1) close convergence in the small dimensional,
> large data regime; and (2) monotonic increase in linear similarity of learned representations as a
> function of dataset size and model capacity in the high dimensional regime."
>
> We appreciate that this was not sufficient for understanding, though, and will add a note to the end of the experimental results section that restates and summarizes the connection. Stated differently from above, we show that the result holds as expected when the dimensionality is relatively small and there is a lot of data, and we can control for other factors. In this situation (figures 1 and 2), the learned embeddings are closely linearly related. Because the embeddings also depend on how well the model is optimized, this becomes more challenging in higher dimensions, and so we instead show tendency towards the linear relationship that Theorem 1 asserts. With better optimization and increasing amounts of data, we expect that our result will become more and more practical in the future. At present, we can only prove that it is true and provide empirical support for it.
>
> >I disagree with the authors claim that “these experiments validate Theorem 1”
>
> We agree that “validate” is a poor word choice here. Theorem 1 is true by the proof we presented in the main body and in the Appendix, regardless of the empirical results. We will change this to “these experiments provide support for the result in the practical regime,” and will add additional discussion in the rebuttal draft.

---

> > ### Comment · AnonReviewer3 · 2020-11-17
> > **feedback on responses**
> >
> > Dear Authors,
> > Thank you very much for providing detailed responses.
> > I see you are still working on a revising version of the paper correcting some issues and improving some parts. I look forward to reading the new version when it is ready.

---

> ### Author Response · Authors · 2020-11-17
> **Thank you for the careful reading and criticism; comments requested (1 of 2)**
>
> Thank you kindly for the careful reading and astute constructive criticisms of this work. We are incorporating these improvements into the rebuttal draft that we will resubmit. We have answered the questions in the comments below, and would appreciate any additional comments about our responses or the existing materials in the paper while we improve the submission.
>
> >Figure 1: This figure is not referenced in the text. It should be referenced as a motivating fact.
>
> Thank you for noticing this omission in the main text. A reference to Fig. 1 in the introduction appears to have been eliminated during the final draft erroneously. This will be corrected in the rebuttal draft.
>
> >Figure 1: It is mentioned in the Appendix that the subset of words was selected. How these subsets were chosen? Randomly? I think this is important to clarify.
>
> The subsets were chosen completely at random. We will add a remark to this effect to the experimental results section. We provided the code for this example so that interested readers could investigate these issues themselves, but agree that it would be inconvenient to have to read code to answer straightforward methodological questions and will fix this.
>
> >Theoretical claims: This figure clearly illustrate a strong relationship between f’ and f*, which is close to a linear mapping, but it is not exactly linear. The main result of the paper is that if diversity condition is met, then models are linearly identifiable, but nothing is said about approximately linearly identifiable models as the ones shown in Figure 1. One question that should be addressed is how a perturbation in the diversity condition is translated to the identifiability property. It would be good that the authors add some theory about it.
>
> Yes, the mapping in Figure 1 is close to linear but is not exact. The model used in Figure 1 is linearly identifiable, a fact that we show in Appendix D (see the last page). We would like to underscore that linear identifiability is a binary property of a statistical model: either the model has it or it does not. When the diversity condition is met, and the model can be reduced to the canonical form of eq. (1),  then the model has the property. This is a different question as to whether particular representations of input data learned by the model lie within an exact linear transformation of each other. This latter question depends on how well optimized the particular instance of the model is. This may seem restrictive, but the regime of large data and improving optimization is increasingly the regime of practice in representational learning for deep learning.
>
> >It is suggested that eq. (1) holds for arbitrary supervised learning problem but, in the Appendix, the equivalence is not shown for this case.  Is it possible to write any supervised learning problem as eq. (1)? Could you please include its derivation?
>
> With apologies, we removed the derivation from the Appendix and only retained plain English explanations in the text. We will add this back in to avoid confusion. Any supervised classification problem can be written as eq. (1), but not every such model will be linearly identifiable because it must meet the diversity conditions. We stress that our goal here was to discuss linearly identifiable learned representations, rather than classifiers.
>
> >Also, the example provided in Fig. 2 seems to be a very arbitrary case.
>
> Yes, Figure 2 was chosen to exactly meet the conditions of the proof, in a somewhat arbitrary fashion (as we say in the text, the 18 classes were intended to match the K >= M+1 case). This synthetic example is only meant to verify that when we control all factors, linear identifiability as in Theorem 1 is an evident property of the learned representations.
>
> >Is the form of eq. (1) already used as a general approach for different ML problems as the authors claim? If so, could you please cite relevant references?
>
> In Section 4 and Appendix D we cite a number of different ML problems for which eq. (1) is a general approach. We also cite particular models that have this property in the introduction. In general, pre-training representations for deep neural networks, which is the focus of our theory, has often used the form of eq. (1). We show these reductions mathematically in Appendix D. Do these materials satisfy your request for citations?

---

### Official Review · AnonReviewer4 · 2020-10-28

**Rating:** 7
**Confidence:** 3

**Review:**

This paper investigated the identifiability of the learned representations in pre-trained DNN models that fall into a general class of function space,  defined by the canonical mathematical form.
The identifiability of learned representations, in this paper, is defined as the representations are reproducible on the same data
distribution, regardless of the randomness in the training procedure such as the random initialization of parameters and the stochastic optimization procedure.

The authors first proved that, in the limit of infinite data, learned representations in this family are asymptotic identifiable in function space up to a linear transformation, with the additional assumption of the diverse condition.

They further showed that this property is applicable to several state-of-the-art pre-trained models, including CTC, BERT and GPT-2 and GPT-3.

At last, the authors conducted experiments to empirically investigate the linear identifiability of DNN models in a practical setting: finite data and partial optimization.  They adapted Canonical Correlation Analysis (CCA) (and SVCCA for high-dimensional space) to measure the linear similarity between two learned representations. Results from three sets of experiments, including classification, self-supervised learning for images (CTC) and for texts (GPT-2). Results show that the learned representations, after mapping through the optimal linear transformation from CCA, have a strongly linear relationship.

Overall, this paper is well-written and well-motivated. From the theoretical aspect, this paper proved the linear identifiability of a large class of DNN models in an ideal setting. From the empirical aspect, they provided experimental results to demonstrate the theorem in a practical setting.

---

> ### Author Response · Authors · 2020-11-17
> **Thanks for your time and the review**
>
> We appreciate the time you spent reviewing and your careful attention to and comments on the paper. We are glad you found the result interesting, and hope that you find the theorem useful in framing your own research topics.

---

> > ### Comment · AnonReviewer4 · 2020-11-18
> > **Feedback on Response**
> >
> > Dear Authors, Thank you very much for providing detailed responses. I have also read your responses to other reviewers.
> >
> > I decide to keep my initial score and would like to see the submission to be accepted.

---

### Official Review · AnonReviewer2 · 2020-10-29
**Valuable results but a longer version would be more suitable. Important details missing.**

**Rating:** 4
**Confidence:** 4

**Review:**

%%% post-rebuttal %%

The authors replied to my comments related to the diversity condition in 3.2 and Theorem 1. Their answers did not fully clarify my concerns or misunderstandings, and it seems the authors didn't make any changes in that regard in the revised version. Except if I missed something in the review system, they did not answer my other comments.

%%%%%%%%%%%%%%%


The paper discusses identifiability of learnt representations in supervised settings and under "canonical discriminative" modelling. It provides conditions under which two learnt representations are equivalent up to a linear transform. The paper discusses an important topic (related in particular to explicability of DNNs) and appears to provide valuable results but is difficult to follow. The paper is packed with results and difficult to read as important details are missing. A longer journal version would be more appropriate.

Comments:

- I got lost from the paragraph "Diversity condition". I don't understand the meaning of y_A and y_B. What do subscripts A and B refer to ? Then what do y_A^(i) and y_B^(i) refer to ? What are they exactly samples of ? Using illustrative examples would be very useful there. For example, what do they mean in classification where y is a label ?

- Because I didn't understand the meaning of y_A and y_B, I failed to understand the impact of Theorem 1. In particular, I don't understand why Eq. (5) holds in the proof.

- I am mildly familiar with the "canonical discriminative" formulation. Can all DNN tasks but represented as such ? How does it relate to more traditional forms of deep learning, such as based on minimisation of cross entropy or quadratic loss ? I assume that the sum in Eq. (1) is an integral when y is continuous.

- I got the idea of using CCA for comparing two different representations but the exact definition of C_i is unclear. Is it a vector or matrix ? Besides, why using only a subset B of D ?

- In the experiments you basically compare two learnt representations and show they are essentially similar up to a linear transform. However it's not clear how the two representations have been obtained. Is it only a matter of different initialisations ?

- I didn't understand how are the labels constructed in Section 5.1. Do you simply divide the unit circle in K=18 vectors that form the mean of each cluster ? I didn't get the "model misspecification" argument and the need to use DNN on such a simple task.

- "This show that increasing model size correlates strongly with increase in linear similarity of learned representations." I understand why this can be true when increasing number of samples but I don't understand why this should be expected with model capacity. What is it in Theorem 1 that reflects this ?


Minor:
- homogenise use of "Equation (x)" and "equation (x)"
- missing compiled reference at the beginning of Section 4
- it looks like you are using two different fonts for f_theta in the paper. Or are those different variables ?
- many typos: "share parameters", "but OF the network", "a the", ...

---

> ### Author Response · Authors · 2020-11-17
> **Many thanks for detailed feedback and request for further comment (2 of 2)**
>
>
> >>> Because I didn't understand the meaning of y_A and y_B, I failed to understand the impact of Theorem 1. In particular, I don't understand why Eq. (5) holds in the proof.
>
> Equation (5) holds based on the definitions in Section 2 and from the assumption in the statement to be proved.  We hope that we have made this clearer through our explanations above. The impact of Theorem 1 is to tell us what is true for pairs of parameters in the model family defined in Section 2 when the data is sufficiently diverse (which is the diversity condition, which tells us when the matrices L* and L’ in the proof are invertible). When these conditions hold, Theorem 1 establishes that the learned representations are equal up to a linear transformation.
>
> (Comment 2 of 2)

---

> ### Author Response · Authors · 2020-11-17
> **Many thanks for detailed feedback and request for further comment (1 of 2)**
>
> We thank you for your time and efforts in reviewing our paper and providing constructive feedback. We appreciate the many comments you provided.
>
> We regret that you found important details to be missing. We took great care to describe all these symbols in Sections 2 and 3. However, because you found these confusing during the proof, and to aid clarity, we will add more elaboration in the text in a coming revision as well as include a brief section on notation immediately before the diversity condition and proof that restates and refreshes the key details.
>
> In the following, we answer your key questions with reference to the text and also provide additional explanation. We hope this clears up your understanding, and that our proposed edits will clear things up for future readers. As these misunderstandings were at the heart of your review, we hope you consider raising your scores, or help us understand what we can do to further increase clarity.
>
> >>> “I got lost from the paragraph "Diversity condition". I don't understand the meaning of y_A and y_B.”
>
> In the first sentence of the Diversity Condition in Section 3, we state that for any datapoint x, the set S is a sample from p(x|S), and that the target variables y_A and y_B are elements of the set S. The meaning of y_A and y_B is stated informally in English and formally in standard mathematical notation in Section 2.  The random variable y and S are related by joint distribution along with an input variable x. This joint distribution is our model of some real world data. In the bullet points of Section 2, we write that y is a target random variable, such as a class label, and S is a set that contains each possible value of y conditioned on a value of x.
>
> >>> What do subscripts A and B refer to ?
>
> A and B are identifiers that we chose to distinguish distinct realizations of the random variable y that are collected together in a set S. In other words, y_A and y_B are realizations of the random variable y, as well as being distinct elements of the set S, necessitating a subscript to show that they are different.  In the same line as y_A and y_B, S is defined as a sample from p(S|x) for a particular input variable x. The random variable S is defined in Section 2 as a set containing the possible values of the target variable y given the input variable x.
>
> >>> Then what do y_A^(i) and y_B^(i) refer to ?
>
> The symbols “y_A^(i), y_B^(i)” appear within the 2-tuples in the set {(y_A^(i), y_B^(i))}_{i=1}^M. The index i indicates that the set contains M different 2-tuples, where A and B are being still being used to denote distinct samples from some S that we have paired together in the 2-tuple. In the proof, we later use the distinct M pairs y_A and y_B to construct M difference vectors d_i = g(y_A^(i)) - g(y_B^(i)). These M different difference vectors d_i must be distinct because they are collected together as columns of the very linear transformation that relates any two learned representation functions f’ and f*. The indices i as well as A and B together ensure that the columns are distinct and linearly independent, under the diversity condition.
>
> >>> What are they exactly samples of ? Using illustrative examples would be very useful there.
>
> We give the following concrete examples of y and S in Section 2: “for supervised classification, S is deterministic and contains class labels. For self-supervised pretraining, S contains randomly-sampled high-dimensional variables such as image embeddings. For deep metric learning (Hoffer and Ailon, 2015; Sohn, 2016), the set S contains one positive and k negative samples of the class x is associated with in the training data.”
>
> >>> “ For example, what do they mean in classification where y is a label?””
>
> As we wrote in the quoted text from Section 2, S contains class labels in the case of classification, and y_A and y_B are distinct class labels. We note in the quoted text that S is deterministic and not random in the case of classification, meaning that p(S|x) is a Dirac delta distribution on a particular S, the one containing the class labels for the classification problem at hand.  In Section 4, we give other illustrative examples with citations to previous work. In Appendix D, we show that these are equivalent mathematically to Equation (1). For generality, we must allow different, stochastic S for different realizations of x.
>
> In Section 4, we elaborate on the classification example: “Classification models that deploy DNNs as feature extractors satisfy the sufficient conditions given in Section 3 when: the network from input to the layer prior to the logits is the representation function f(x) and weights in the final projection layer are given by a context map g(y) that depends only on the labels. The class labels y are integers 0 to K 1, and wi = g(y = i) is the i-th column of a weight matrix W, and the vector representation of the i-th class.”
>
> (Comment 1 of 2)

---

### Official Review · AnonReviewer1 · 2020-10-30
**Seems like a good paper, but not clear to me what are the practical benefits**

**Rating:** 6
**Confidence:** 3

**Review:**

This paper claims that, through Canonical Correlation Analysis on the representations learnt by popular deep models, we can show that the representations learnt on the same dataset, with same model using different sets of parameters, the representations are linearly identifiable.

This seems like a nice thing to know. But I however am not sure about few things.
1) I think the paper doesn't really discuss the implications of this enough. That is, I would like to see few example applications where we take advantage of this fact. As is, this contribution in my opinion remains as a cute thing to know.
2) It would be interesting to how much would change in initializations would effect the CCA curves. That is, if I initialize my network within a wider range, the identifiability result would still hold in practice? This would be interesting to add in my opinion.
3) Ideally I would like to see if this statement hold for different hyperparameters also.

Overall, the conclusion is interesting and I think that it could be published.

---

> ### Author Response · Authors · 2020-11-17
> **Answers to questions on empirical robusts and applications will be added to paper; request for comment**
>
> Thank you for the time and efforts given to reviewing our paper. Your questions are important for understanding the present and future impact of our result and we appreciate the opportunity they give us to address them in our rebuttal.
>
> We respond below to your questions. We kindly request that you briefly read these responses and let us know if they do not fully satisfy your concerns and curiosity about the result or its implications and potential applications.
>
> >I think the paper doesn't really discuss the implications of this enough. That is, I would like to see few example applications where we take advantage of this fact. As is, this contribution in my opinion remains as a cute thing to know.
>
> We agree that more discussion on implications and practical benefits would be helpful and appreciate the suggestion. We have added the following paragraph to the paper:
>
>  “Identifiability results are (as we aim to demonstrate) helpful in predicting when learned optimal representations are easily reproducible or not. While non-identifiable models might need to be trained many times to reproduce optimal results, identifiable architectures would (in principle) only need to be trained once, reducing the amount of computational resources required, thereby reducing energy waste, as well as saving budget and labour time. Moreover, the field of deep learning is characterized by increasingly efficient function approximators trained on growing quantities of data. This trend interfaces with the asymptotic character of Theorem 1 and linear identifiability in a productive way: as network architecture design and optimization continues to improve and more data is collected, we expect the representation functions approximated by deep neural networks to approach a stable set of optima in function space.”
>
> As a side note, representation learning is the central topic of this paper, and so learning disentangled representations and causal representations for, e.g., medical applications are also interesting applications. Due to space constraints we did not explore these in detail, but will add in comments. Our goal in this paper was to introduce the idea so that practical applications can follow on a solid theoretical basis.
>
> >It would be interesting to how much would change in initializations would effect the CCA curves. That is, if I initialize my network within a wider range, the identifiability result would still hold in practice? This would be interesting to add in my opinion.
>
> Thanks for this interesting question! We will add a discussion section to the end of the empirical results section to address this point. Whether the curves look the same depends on how the models are optimized: if the optimization routines (like Adam, AdaGrad, etc.) are robust to wider initialization within a certain range, we can expect the curves to look similar but take longer to converge. However, this cannot make up for poor initialization or poor optimization, however: just as in any deep neural network, a poor initialization and inadequate optimizer will interfere with learning the model parameters. In the case of a linearly identifiable model, means that the learned representations would not appear to be within a linear transformation of each other, since the models have failed to converge due to poor initialization.
>
> We would like to note that all models were initialized using standard techniques in deep learning, and so our results are representative of the typical use case of deep neural networks for representation learning. We will add this remark to the paper as well.
>
> >Ideally I would like to see if this statement hold for different hyperparameters also.
>
> Thanks also for this interesting question! We will add a variation of the following answer to the aforementioned discussion section in the rebuttal draft: “As the hyperparameters of a model are changed, this changes what class of functions the network could learn (i.e., the size and stride of convolution filters will change what input pixels could be correlated in deeper layers). Typically in deep learning models, the hyperparameters are carefully tuned using cross validation based on held-out data. We did so in our experiments also. We expect that such a tuning procedure would yield hyperparameters that are as good as possible for the model to be optimized, allowing sufficient optimization so that the linear identifiability of the learned representations is realized. If the hyperparameters are sufficiently bad and optimization suffers, this will interfere with linear identifiability of the learned representations also.”
>
> We hope the above answers your questions!

---

### Author Response · Authors · 2020-11-25
**Thank you for your patience. Rebuttal draft complete!**

Dear reviewers,

Thanks for your patience. We have submitted an extensive rebuttal redraft that incorporates the valuable feedback received from each reviewer. This has increased the length of the submission to 9 pages and added substantial materials to the Appendix.

We hope you find these changes to be a strong improvement from the previous draft, thanks to your constructive and detailed feedback.

---

### Decision · Program_Chairs · 2021-01-07
**Final Decision**

**Decision:**

Reject

**Comment:**

This paper presents novel results on linear identifiability in discriminative models, with three of the four reviewers arguing for acceptance. The paper went through an extensive round of edits, which incorporated detailed responses to issues raised by the reviewers.

While this paper would be a nice contribution to the conference, some reviewer concerns remain unresolved, so we encourage the authors to revise and resubmit to a future venue.